# Selective Annotation Makes Language Models Better Few-Shot Learners

**Hongjin Su**♠  **Jungo Kasai**♣◇  **Chen Henry Wu**♡  **Weijia Shi**♣  **Tianlu Wang**♦  **Jiayi Xin**♠
**Rui Zhang**★  **Mari Ostendorf**♣  **Luke Zettlemoyer**♣♦  **Noah A. Smith**♣◇  **Tao Yu**♠♣
♠The University of Hong Kong  ♣University of Washington  ◇Allen Institute for AI
♡Carnegie Mellon University  ★Penn State University  ♦Meta AI

{hjsu,tyu}@cs.hku.hk, henrychenwu@cmu.edu, ostendor@uw.edu
{jkasai,swj0419,lsz,nasmith}@cs.washington.edu

## Abstract

Many recent approaches to natural language tasks are built on the remarkable abilities of large language models. Large language models can perform in-context learning, where they learn a new task from a few task demonstrations, without any parameter updates. This work examines the implications of in-context learning for the creation of datasets for new natural language tasks. Departing from recent in-context learning methods, we formulate an annotation-efficient, two-step framework: *selective annotation* that chooses a pool of examples to annotate from *unlabeled* data in advance, followed by prompt retrieval that retrieves task examples from the annotated pool at test time. Based on this framework, we propose an unsupervised, graph-based selective annotation method, vote-$k$, to select diverse, representative examples to annotate. Extensive experiments on 10 datasets (covering classification, commonsense reasoning, dialogue, and text/code generation) demonstrate that our selective annotation method improves the task performance by a large margin. On average, vote-$k$ achieves a 12.9%/11.4% relative gain under an annotation budget of 18/100, as compared to randomly selecting examples to annotate. Compared to state-of-the-art supervised finetuning approaches, it yields similar performance with 10-100× less annotation cost across 10 tasks. We further analyze the effectiveness of our framework in various scenarios: language models with varying sizes, alternative selective annotation methods, and cases where there is a test data domain shift. We hope that our studies will serve as a basis for data annotations as large language models are increasingly applied to new tasks.[1]

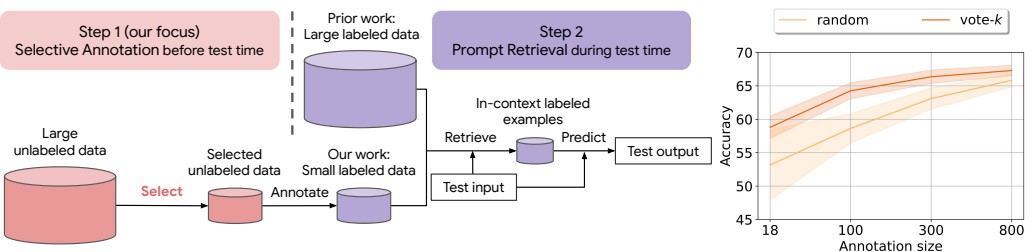

Figure 1: **Left**: Our two-step framework for in-context learning. Instead of assuming access to large labeled data, we first select a small number of (diverse and representative) unlabeled examples to annotate before test time. At test time, we retrieve in-context examples from the small annotated pool. **Right**: In-context learning performance over varying annotation budgets averaged over three representative tasks (HellaSwag commonsense reasoning, MRPC paraphrase detection, and MWOZ dialogue state tracking). Here we experiment with GPT-J and Codex-davinci-002. Two selective annotation methods are presented: *random selection* and our *vote-$k$* method. We observe that an appropriate selective annotation method largely improves the in-context learning performance with smaller variance over random selection under varying annotation budgets.

---

[1]Our code is available at https://github.com/HKUNLP/icl-selective-annotation.

# 1 INTRODUCTION

Much recent work builds approaches to natural language tasks on the impressive abilities of large language models (e.g., GPT-3; Brown et al., 2020). Large language models can perform downstream tasks by conditioning generation on a few task demonstrations, thereby avoiding the need for any parameter updates. This new, few-shot learning paradigm is called *in-context learning* and has become an attractive alternative to supervised finetuning (Liu et al., 2021). In this work, we study the implications of this remarkable capability of large language models for dataset creation and annotation. We extensively examine how to reduce the manual annotation cost while retaining high in-context learning performance.

Although in-context learning was originally proposed for few-shot learning, recent works show that retrieving prompts from a large set of annotated examples is necessary to achieve good performances (Liu et al., 2022; Rubin et al., 2022). In particular, they show that the performance substantially improves when similar examples (under some embedding function) are retrieved as in-context examples specifically for each test input (Liu et al., 2022). Each test sample only requires a few in-context examples in its prompt. Different test instances, however, require different in-context examples with their associated annotations, necessitating a large set of annotated examples.

Distinct from these recent efforts, we establish a two-step framework to better understand and improve the annotation efficiency (Fig. 1): the first step is *selective annotation* that picks a small number of instances to get annotated before test time, followed by *prompt retrieval* that retrieves in-context examples for each test instance from the annotated data. The total annotation budget is the number of examples selected and annotated in the first step. The second step is bounded by the number of examples that can fit as input to a language model. Based on this framework, we propose an unsupervised, graph-based selective annotation method, named vote-$k$, that selects diverse and representative instances to be annotated.

Our extensive experiments over 10 datasets across diverse tasks (covering classification, commonsense reasoning, dialogue, and text/code generation; see Tab. 2) demonstrate that our graph-based selective annotation method, vote-$k$ (§2.1), substantially improves the in-context learning performance by balancing the diversity and representativeness of annotated samples. For instance, vote-$k$, combined with similarity-based prompt retrieval (Liu et al., 2022; Rubin et al., 2022), achieves a 11.4% relative gain under a budget of 100 annotated examples and a 12.9% relative gain when only 18 examples are annotated; 18 samples can fit into language models' input, meaning the prompt retrieval step is not needed. Moreover, the improvement is consistent across language models with varying sizes (2B-175B parameters) (§4.2). This finding is in contrast with finetuning, where we cannot see the effectiveness of selective annotation over random baseline, due to outliers (Karamcheti et al., 2021) or training instability (D'Arcy & Downey, 2022). We hypothesize that in-context learning *with similarity-based prompt retrieval* is more robust to small annotation sizes and outliers because only the most similar examples are retrieved for each test instance. Indeed, we observe that *random prompt retrieval* fails to benefit from selective annotation (§4.4), providing support for our hypothesis.

Besides performance comparisons within a fixed annotation budget, we show that selective annotation provides better few-shot performance with 5-100× *less annotation cost* for new natural language tasks. In-context learning with 18 examples selected by vote-$k$ achieves higher performance than 100 randomly selected examples on 6 out of the 10 tasks. It also outperforms strong finetuning methods by a large margin (Fig. 2) and requires 10-100× less annotations for similar performance (§4.1). We observe that in-context learning quickly (100 or 300 samples are annotated) converges to decent performance when vote-$k$ selective annotation is applied. These results suggest that large language models do not require large annotated datasets (e.g., 10K) due to their ability to adapt to new tasks through simple prompting.

Selective annotation also makes in-context learning much more *stable*. In real-world scenarios, even collecting *unlabeled* data is non-trivial and introduces randomness. We simulate such randomness in our experimental setting by subsampling the original unlabeled data multiple times. Our results suggest that vote-$k$ selective annotation largely reduces the variance of in-context learning even in this setting (Tab. 2). Further analysis shows larger improvements when there is a domain shift between training and test data (e.g., text from different Amazon users; Koh et al., 2021; §4.3). Finally, when compared with previous selective annotation methods designed for supervised training/finetuning, we demonstrate that vote-$k$ selective annotation consistently improves the performance (§4.5). As

in-context learning has been applied to increasingly more natural language processing applications, we hope that our annotation-efficient framework will provide useful guidance for both researchers and practitioners.

## 2   SELECTIVE ANNOTATION FOR IN-CONTEXT LEARNING

In-context learning only requires a few annotated examples per test instance (*few-shot learning*), while avoiding expensive finetuning on the whole training data. It is, however, often assumed that all *annotated* training data are available for prompt retrieval (e.g., Liu et al., 2022; Rubin et al., 2022). Yet the implied total annotation costs are hardly discussed in previous work. We develop a better practice for few-shot learning with large language models by carefully studying the total annotation cost required for in-context learning. We also study how examples should be selected to annotate, in order to make in-context learning perform better for new tasks. We formulate a general framework (Fig. 1 left) that consists of two steps: selective annotation (§2.1) and prompt retrieval (§2.2).

### 2.1   SELECTIVE ANNOTATION

The first step chooses examples to annotate *before* test time. This process thus determines the total annotation budget. This selective annotation process is largely ignored in the recent literature for in-context learning. We will demonstrate, however, that the annotation cost can be substantially reduced by choosing a small set of diverse, representative examples, while retaining the downstream performance (§3). Formally, given a set of unlabeled samples $\mathcal{X} = \{x_i\}_{i=1}^{N}$, selective annotation aims at selecting a subset $\mathcal{L} \subset \mathcal{X}$ to be annotated, where $|\mathcal{L}| = M$ is the annotation budget. We discuss our vote-$k$ selective annotation method and other selective annotation baselines below.

**Vote-$k$**   The goal of selective annotation for in-context learning is to select diverse and representative examples; representativeness will help many test instances to find similar demonstrations, while diversity increases the total coverage. We develop vote-$k$, a graph-based method that promotes both diversity and representativeness. A detailed algorithm can be found in Appendix G. We first compute a vector representation for each *unlabeled* training instance using Sentence-BERT (Reimers & Gurevych, 2019) by averaging the resulting vectors over the text input words.[2] We then use the embedding vectors to create a directed graph $G = (V, E)$ where the vertices $V$ are the unlabeled instances $\mathcal{X}$ as defined above. For each vertex $v \in V$, we create an edge to its $k$ nearest vertices in terms of the cosine similarity between the embeddings. Now let $\mathcal{L}$ and $\mathcal{U}$ denote the sets of already chosen (i.e., labeled) samples and remaining samples, respectively. Initially, $\mathcal{L} = \emptyset$. Every vertex $u \in \mathcal{U}$ is scored by a modified degree:

$$\text{score}(u) = \sum_{v \in \{v|(v,u) \in E, v \in \mathcal{U}\}} s(v), \quad \text{where } s(v) = \rho^{-|\{\ell \in \mathcal{L}|(v,\ell) \in E\}|}, \quad \rho > 1$$

where $s$ discounts $v$ that is close to the already selected instances, thereby encouraging diversity. In every iteration, we take $\arg\max_{u \in \mathcal{U}} \text{score}(u)$ and move it from $\mathcal{U}$ to $\mathcal{L}$. We run $M/10$ of these iterations; after this process, the current labeled $\mathcal{L}$ has $M/10$ samples (up to Line 7 in Algorithm 1). Subsequently, we use $\mathcal{L}$ as the in-context learning examples for large language model, e.g.,GPT-J (Wang & Komatsuzaki, 2021), and generate a prediction for every instance in $\mathcal{U}$. We then compute the average log probability over the generation output as the model's confidence score (Line 8 to Line 10 in Algorithm 1). We then partition $\mathcal{U}$ into $M$ equal-sized buckets, based on their confidence scores (e.g., if $M = 100$, we group the unlabeled instances by percentile). We add to $\mathcal{L}$ the example with the maximum score from each of the first $9M/10$ buckets (discarding the $M/10$ buckets with the most confident examples), resulting in $|\mathcal{L}| = M$ (Line 11 to Line 15 in Algorithm 1). This further encourages diversity by selecting instances with varying confidence scores from in-context learning. We tuned $k$ and $\rho$ in our preliminary experiments, and found that $k = 150$ and $\rho = 10$ perform well across many datasets.[3] We will explore other selective annotation methods from prior work on active learning or coreset selection (§4.5) and see that vote-$k$ outperforms these alternative methods.

**Random and Other Selective Annotation Methods**   To quantify the effect of selective annotation, we also provide random and other baselines. For randomly-selected annotation, we conduct

---

[2]https://huggingface.co/sentence-transformers/all-mpnet-base-v2.

[3]We also explored $k = 100, 150, 200$, but it did not lead to substantial performance changes.

experiments three times and report the average score. We will show that these baselines substantially underperform the vote-$k$ method (§3.3), demonstrating the importance of the selective annotation step to reduce the total annotation cost.

## 2.2 PROMPT RETRIEVAL

Once we have a set of annotated examples $\mathcal{L}$ from selective annotation, we retrieve a few examples from the annotated set as in-context examples for each test instance. Following recent work (Liu et al., 2022), we will compute embeddings for all annotated samples using Sentence-BERT and find the most similar examples to each test instance in terms of cosine similarity.

## 3 EXPERIMENTS

We conduct extensive experiments over 10 diverse datasets, spanning 9 distinct tasks, and show a better approach to few-shot learning than previously considered. In general, we find that the first step of selective annotation is particularly crucial to reduce the amount of required annotation.

## 3.1 DATASETS AND TASKS

We use 10 diverse NLP datasets across 9 tasks that are listed in Table 1. These datasets involve different task formulations, thereby allowing for extensive evaluations in varying scenarios. Some of those are included in the widely-used GLUE benchmark (Wang et al., 2019). Appendix A illustrates details of the 10 datasets with examples.

For each dataset, we use the standard train/dev./test split available from the Transformers library (Wolf et al., 2020). In the selective annotation step, we remove all labels in the training data. For the datasets that have test data available publicly, we use the the test data for evaluation (SST-5, XSUM, MWoZ, and DBpedia). For the others, we follow prior work (e.g., Jiang et al., 2020; Lan et al., 2020; Gao et al., 2021) and use the dev. data for evaluation.[4] We evaluate the methods by accuracy for all classification and multiple-choice selection datasets, joint accuracy (Budzianowski et al., 2018) for MWoZ, test suite accuracy (Zhong et al., 2020) for GeoQuery, exact matching (Rajpurkar et al., 2016) for NQ, and ROUGE-L (Lin, 2004) for XSum.

| | Dataset | Task | In-Context Learning Models |
|---|---|---|---|
| **Classification** | MRPC (Dolan et al., 2004) | Paraphrase Detection | GPT-Neo, GPT-J, GPT-3 |
| | SST-5 (Socher et al., 2013) | Sentiment Analysis | GPT-J |
| | DBpedia (Lehmann et al., 2015) | Topic Classification | GPT-J |
| | MNLI (Williams et al., 2018) | Natural Language Inference | GPT-J |
| | RTE (Bentivogli et al., 2009) | Natural Language Inference | GPT-J |
| **Multiple-Choice** | HellaSwag (Zellers et al., 2019) | Commonsense Reasoning | OPT, GPT-Neo, GPT-J, GPT-3 |
| **Dialogue** | MWoZ 2.4 (Budzianowski et al., 2018) | Dialogue State Tracking | Codex-{cushman, davinci-002} |
| **Generation** | GeoQuery (Zelle & Mooney, 1996) | Semantic Parsing | Codex-davinci-002 |
| | NQ (Kwiatkowski et al., 2019) | Open-Domain QA | Codex-davinci-002 |
| | XSUM (Narayan et al., 2018) | Summarization | GPT-J |

Table 1: All the 10 datasets and the in-context learning models used in our experiments. GPT-J and Codex-davinci-002 are used by default. Other in-context learning models are explored in analysis.

**Measuring Stability**   Given a set of unlabeled data, our vote-$k$ selective annotation algorithm is *deterministic*, without any randomness. However, we note that in real scenarios, even getting *unlabeled* samples is not trivial, and getting unlabeled samples can be a process with large variance. To simulate this real setting, we perform selective annotation from 3K instances that are randomly subsampled from the original training data for each task. For each experiment, we repeat this subsampling three times, and results are averaged over the three trials. We will find that vote-$k$ still substantially improves stability over alternative selective annotation methods.

---

[4]The one exception is GeoQuery, where we concatenated the dev. and test data to have reliable evaluations on larger data.

## 3.2 IN-CONTEXT LEARNING MODELS

We mainly perform experiments using GPT-J with 6B parameters (Wang & Komatsuzaki, 2021) due to our computational budget. The exceptions are the MWoZ, GeoQuery, and NQ datasets, where we use Codex-davinci-002 (Chen et al., 2021),[5] a variant of GPT-3 finetuned on code data from the web. Codex is particularly effective for structured prediction such as semantic parsing, and we found it is indeed effective on three datasets (MWoZ, GeoQuery, and NQ) in our preliminary experiments. We will explore the effectiveness of selective annotation on the largest publically available language models, OPT-175B (Zhang et al., 2022) for HellaSwag (Fig. 4) and Codex-davinci-002 for MWoZ, over varying annotation budgets. We will also explore other language models with different sizes for three representative tasks (HellaSwag, MWoZ, and SST-5) in §4.2: GPT-3 with 175B (Brown et al., 2020) and GPT-Neo with 2.7B parameters (Black et al., 2021). Our later experiments will show the same patterns among selective annotation methods over these different language models. For the classification and multiple-choice tasks, we compute the average log score for each choice and choose the maximum one. For generation tasks, we simply perform beam-search decoding.

See Appendix B for our in-context learning prompt templates for all 10 datasets. For every test instance, we feed as much retrieved samples as possible into the language model until the maximum token length is reached. On average, the number of samples $N$ fed into the language model is 13.4 across different experiments. The in-context examples are concatenated in the ascending order of the similarity so that more similar examples benefit from the recency bias (Lu et al., 2022).

## 3.3 MAIN RESULTS

| Method | | Classification | | | | | Multi-Choice | Dialogue | Generation | | |
|---|---|---|---|---|---|---|---|---|---|---|---|
| $\mathcal{L}$ | Selection | MRPC | SST-5 | MNLI | DBpedia | RTE | HSwag | MWoZ | GeoQ | NQ | XSum |
| 100 | Random | 63.5 | 44.2 | 37.4 | 89.8 | 51.5 | 65.2 | 47.2 | 78.6 | 30.8 | 15.3 |
| 100 | Vote-$k$ | **70.7** | **53.0** | **47.3** | **93.4** | **55.5** | **70.7** | **51.4** | **82.8** | **33.6** | **17.2** |
| 100 | $\Delta$ Absolute gain | +7.2 | +8.8 | +9.9 | +3.6 | +4.0 | +5.5 | +4.2 | +4.2 | +2.8 | +1.9 |
| 18 | Random | 59.6 | 39.8 | 36.7 | 77.6 | 50.4 | 62.5 | 33.6 | 62.4 | 29.8 | 13.6 |
| 18 | Vote-$k$ | **64.2** | **47.6** | **41.0** | **87.1** | **54.3** | **67.4** | **42.8** | **72.5** | **32.3** | **15.2** |
| 18 | $\Delta$ Absolute gain | +4.8 | +7.8 | +4.3 | +9.5 | +3.9 | +4.9 | +8.8 | +9.9 | +2.5 | +1.6 |

Table 2: In-context learning results with randomly-selected and vote-$k$ selective annotation methods on all 10 datasets, with an annotation budget of 100 or 18. There is no prompt retrieval step when only 18 samples are annotated since all annotated samples can fit into the in-context learning model's input. Across the board, selective annotation with vote-$k$ substantially outperforms the randomly-selected annotation baseline for in-context learning. Further, vote-$k$ largely reduces the variance over three trials (see the min and max results in Appendix C), making in-context learning more stable.

Seen in Table 2 are our results from all 10 diverse datasets with the annotation budgets of $|\mathcal{L}| \in \{18, 100\}$. 18 is chosen so that all annotated examples can be fit to the prompt for the language models without prompt retrieval. Over all datasets, vote-$k$ selective annotation outperforms the random baseline by a large margin (5.2% absolute gain and 11.4% relative gain on average) when the annotation budget is 100. Even when only 18 examples are annotated and fixed as the in-context examples for all testing instances (no prompt retrieval step), in-context learning with vote-$k$ still improves the randomly-selected annotation baseline (5.8% absolute gain and 12.9% relative gain on average). Particularly noteworthy is that in-context learning with 18 examples selected by vote-$k$ achieves higher performance than the one with 100 randomly selected examples on 6 out of 10 tasks. Moreover, vote-$k$ is a deterministic selective annotation method, conditioned on a set of unlabeled samples. Therefore, the variance of vote-$k$ comes solely from how the unlabeled samples are collected, largely improving the robustness of in-context learning. We therefore recommend that researchers and practitioners use selective annotation (e.g., our vote-$k$ method) to better benefit from the few-shot learning capability of large language models with stability. Our later experiments will also illustrate that vote-$k$ consistently outperforms alternative selective annotation methods (§4.5).

---

[5]The parameter size of Codex is not officially confirmed, but it is likely to be 175B.

## 4 ANALYSIS

Our extensive experiments demonstrated that selective annotation is important for the success of in-context learning. Here we conduct detailed analysis to provide further guidance for researchers and practitioners of few-shot in-context learning. We analyze selective annotation for in-context learning from a variety of perspectives: comparisons to finetuning methods (§4.1), varying language model sizes (§4.2), test data domain shifts (§4.3), prompt retrieval methods (§4.4), and alternative selective annotation methods (§4.5).

### 4.1 IN-CONTEXT LEARNING VS. FINETUNING

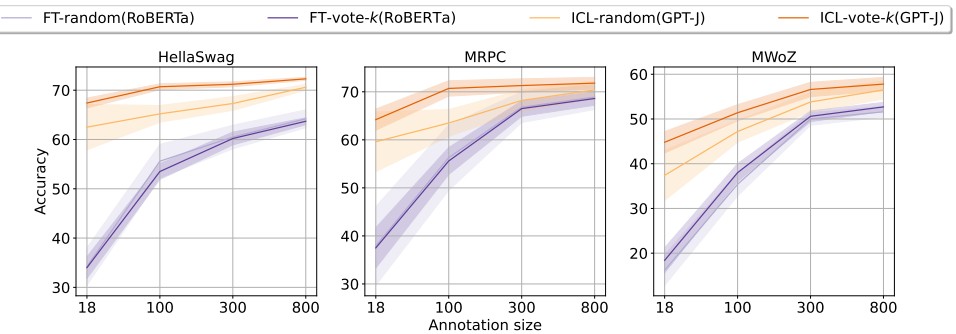

Figure 2: Comparisons between the in-context learning and finetuning paradigms over varying annotation budgets on three representative tasks: HellaSwag commonsene reasoning, MRPC paraphrase detection, and MWoZ dialogue state tracking. Four configurations are presented: finetuning with examples that are randomly selected to annotate (FT-random) or selected by our vote-$k$ selective annotation method (§2.1; FT-vote-$k$) and in-context learning with randomly-selected annotation (ICL-random) or vote-$k$ selection (ICL-vote-$k$). See §4.1 for experimental details. Selective annotation largely improves the in-context learning performance compared to randomly-selected annotation even when the annotation budget is 18. In-context learning with wisely-selected labeled samples is a much better few-shot practice than a strong finetuning method.

As discussed earlier, in-context learning is an alternative learning paradigm to conventional finetuning. Through the lens of our two-step framework, we observed that selective annotation and prompt retrieval are key to the success of in-context learning. A new question now arises: how does in-context learning compare with finetuning under limited annotation budgets? We empirically compare the two paradigms in this section.

We experiment with three representative tasks: MRPC (classification), HellaSwag (multiple-choice), and MWoZ (dialogue). Strong, state-of-the-art pretrained models are used for finetuning: large-sized RoBERTa (Liu et al., 2019) for MRPC and HellaSwag and DS2-T5 (Shin et al., 2022) for MWoZ. In-context learning uses GPT-J for MRPC, GPT-J and OPT 175B (Fig 4) for HellaSwag, and Codex-davinci-002 for MWoZ. Note that we do not aim to conduct head-to-head comparisons with exactly the same pretrained model; finetuning a large left-to-right language model (e.g., GPT-J and GPT-3) is computationally (and thus financially) infeasible in many cases. Here we examine the two paradigms from the practical perspective and benefit from the advantage of in-context learning, which requires no parameter updates of massive language models.

Fig. 2 compares the two paradigms across varying annotation sizes ($\{18, 100, 300, 800\}$). Over all three tasks, we observe that in-context learning with vote-$k$ selection outperforms the finetuning performance of state-of-the-art pretrained language models. Specifically, we find that to achieve similar performance to vote-$k$ with $|\mathcal{L}| = 18$ or 100, finetuning requires 1000 annotated examples for HellaSwag and 800 for MWoZ (**10-100× annotation cost**). Note that the in-context learning performance usually converges when 100 or 300 examples are carefully selected and annotated, suggesting that a large annotated dataset is unnecessary for in-context learning to achieve strong performance. Interestingly, selective annotation helps in-context learning, but *not* finetuning. This result is consistent with recent work showing that many active learning algorithms perform similarly

to random baseline, when pretrained language models are finetuned (Karamcheti et al., 2021; D'Arcy & Downey, 2022). They proposed that it might be due to outliers and the instability of finetuning on a limited number of annotated samples. We hypothesize that in-context learning *with similarity-based prompt retrieval* is more robust to outliers and small annotation sizes because only the most similar examples are retrieved for each test instance. We find two pieces of evidence for this hypothesis. First, § 4.4 shows that *random* (as opposed to similarity-based) prompt retrieval does not benefit from vote-$k$ selective annotation. Second, in Appendix E, we show that explicitly removing outliers also helps finetuning to benefit from vote-$k$.

## 4.2 Language Models with Various Sizes

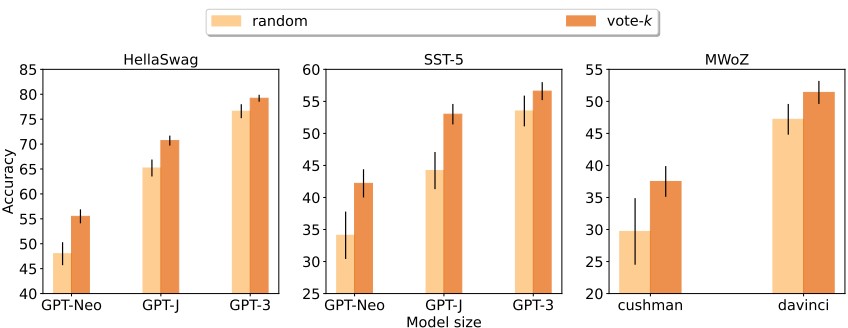

Figure 3: Comparisons of various models with 100 annotated examples. Vote-$k$ selective annotation consistently improves in-context learning with pretrained language models of varying sizes.

Fig. 3 shows performance with varying sizes of language models (GPT-Neo 2B, Black et al., 2021; GPT-J 6B, Wang & Komatsuzaki, 2021; GPT-3, Brown et al., 2020) on HellaSwag commonsense reasoning, SST-5 sentiment analysis, and MWoZ dialogue state tracking. In general, when a smaller model is used, the performance gap between random and vote-$k$ selection is larger. In the HellaSwag task, vote-$k$ outperforms randomly-selected annotation by 7.5% using GPT-Neo, but only 2.6% using GPT-3. Nonetheless, we see consistent performance gains from vote-$k$ selection over varying sizes.

## 4.3 Effects of Domain Shift

Recent work observed that when a large, pretrained language model is finetuned, the performance gain from active learning is limited (D'Arcy & Downey, 2022), but it can be larger if there is a domain shift between training and evaluation (Tamkin et al., 2022). We have demonstrated that selective annotation consistently improves in-context learning, but here we explore cases of domain shifts.

| | **Method** | **CivilComments** | | **Amazon** | |
|---|---|---|---|---|---|
| $|\mathcal{L}|$ | Selection | Random | Domain | Random | Domain |
| 100 | Random | 73.8 | 66.8 | 50.3 | 30.7 |
| 100 | Vote-$k$ | **79.3** | **76.7** | **56.3** | **39.0** |
| 100 | $\Delta$ Absolute gain | +5.5 | +9.9 | +6.0 | +8.3 |

Table 3: Effects of domain shift. Random splits and domain splits are compared (Koh et al., 2021).

Following Tamkin et al. (2022), we use two natural language datasets from the WILDS benchmark (Koh et al., 2021): **CivilComments** (toxicity classification; Borkan et al., 2019) and **Amazon** (review classification; Ni et al., 2019). Each comes with both a random split and a domain split: the former splits data randomly and the latter is based on the domains (demographic identities for CivilComments and users for Amazon), simulating cases where a model is deployed in a new scenario unseen during annotations. Similar to §3.3, we conduct experiments with GPT-J under two settings: random/vote-$k$ selective annotation, followed by similarity-based prompt retrieval. Both selective annotation and prompt retrieval are conducted on the source domain.

Tab. 3 shows our results. We see that the gain from vote-$k$ is more pronounced under the domain splits: e.g., 9.9 vs. 5.5 accuracy point improvements on CivilComments. This suggests that selective annotation and prompt retrieval are particularly crucial when there is a domain shift in the evaluation data, as in many realistic scenarios (Koh et al., 2021; Longpre et al., 2022).

## 4.4 RANDOM PROMPT RETRIEVAL

| | Method | | Dataset | | |
|---|---|---|---|---|---|
| $\|\mathcal{L}\|$ | Selection | Retrieval | HellaSwag | SST-5 | MWoZ |
| 100 | Vote-$k$ | Similar | **70.7** | **53.0** | **51.4** |
| 100 | Random | Similar | 65.2 | 44.2 | 47.2 |
| 100 | Vote-$k$ | Random | 62.5 | 41.6 | 35.6 |
| 100 | Random | Random | 63.2 | 40.6 | 43.8 |

Table 4: Comparison of random and similar prompt retrieval. Random retrieval fails to benefit from diverse and representative annotated examples from vote-$k$.

We have performed similarity-based prompt retrieval so far. Here we experiment with a random baseline for the prompt retrieval step to quantify the effect of prompt retrieval (Tab. 4). Interestingly, when random prompt retrieval is performed, vote-$k$ does not necessarily improve upon the randomly-selected annotation baseline: e.g., 62.5 vs. 63.2 on HellaSwag and 35.7 vs. 43.8 on MWoZ. This suggests that random prompt retrieval fails to benefit from diverse, representative 100 samples, selected by vote-$k$ selective annotation. Combining selective annotation and prompt retrieval is thus crucial for the success of in-context learning.

## 4.5 ALTERNATIVE SELECTIVE ANNOTATION METHODS

| | Random | MFL | K-means | Diversity | Least-conf | Conf-only | Fast vote-$k$ | Vote-$k$ |
|---|---|---|---|---|---|---|---|---|
| HellaSwag | 65.2 | 66.5 | 67.6 | 68.2 | 68.4 | 68.6 | 69.5 | **70.7** |
| SST-5 | 44.2 | 45.6 | 47.2 | 48.5 | 46.2 | 48.3 | 51.9 | **53.0** |
| MWoZ | 47.2 | 48.3 | 48.8 | 49.2 | 49.4 | 49.2 | 50.2 | **51.4** |

Table 5: Comparisons of various selective annotation methods with 100 annotated examples. Performance is averaged over three random trials. Vote-$k$ outperforms all the other selective annotation methods. Fast vote-$k$, a faster version of voke-k without the need for confidence score computations, can achieve similar performance to vote-$k$ while being more computationally efficient.

Here we explore six additional methods for selective annotation:

- **Maximizing facility location** (MFL; Lin & Bilmes, 2009) aims at optimizing the representativeness of the selected samples. Since this objective satisfies the submodular objective, maximization can be approximated via a greedy algorithm (see Appendix G.2).
- **K-means** (Lloyd, 1982) groups examples into $k$ clusters, and selects the centroid example from each cluster.
- **Diversity** focuses on maximizing the diversity of the embeddings for selected examples in the first step (Appendix G.3).
- **Least-conf** (Lewis & Gale, 1994) iteratively adds least-confident examples to the annotated set.
- **Conf-only** is ablations that apply confidence-based stratification similar to vote-$k$, but examples are sampled randomly from each group without graph-based scoring.
- **Fast vote-$k$** is a fast, efficient alternative to our vote-$k$ method (§2.1) that does not use confidence scores. It picks $M$ samples with the largest vote-$k$ scores. It avoids using the pretrained language model to compute a confidence score for every instance, resulting in a 10+ times speedup.

Notice that MFL, diversity, and least-confidence do not have hyperparameters other than the annotation budget. As shown in Tab. 5, vote-$k$ outperforms all the other methods. It is noteworthy,

however, that fast vote-$k$ can achieve similar performance to vote-$k$. Fast vote-$k$ is thus an attractive method for researchers and practitioners with a limited computational budget. Like vote-$k$, MFL also optimizes representativeness and Diversity also optimizes diversity. In particular, MFL defines representativeness as a sum over distances from the selected examples to all other examples, and Diversity defines diversity as the distances between selected examples. Since they do not significantly outperform randomly-selected annotation, we conjecture that jointly optimize diversity and representativeness is needed for selective annotation. Moreover, the way vote-$k$ defines and diversity are also different from the baselines: vote-$k$ defines representativeness as the number of neighbors during similarity-based prompt retrieval, which is effectively tailored to in-context learning; vote-$k$ directly optimizes for the diversity of selected samples using the in-context learning model's prediction confidence.

## 5 RELATED WORK

**In-Context Learning** In-context learning with large language models has recently received an increasing amount of interest, partly due to its flexibility and sample efficiency (Liu et al., 2021). Several recent works proposed methods to improve in-context learning in many aspects: e.g., meta-training (Chen et al., 2022; Min et al., 2022b), task instructions (Efrat & Levy, 2020; Mishra et al., 2022; Wei et al., 2021; Sanh et al., 2022), or task formulation (Holtzman et al., 2021; Zhao et al., 2021; Min et al., 2022a). In this paradigm, the choice of in-context (i.e., demonstration) examples has been shown crucial (Liu et al., 2022; Rubin et al., 2022; Lu et al., 2022), while recent work raised questions as to the degree to which correct labels are necessary (Min et al., 2022c). This work proposes an annotation-efficient in-context learning framework by focusing on the choice of examples and its implications on the annotation cost.

**Active Learning** Active learning aims to enable machine learning models to achieve similar or greater performance with fewer labeled training instances (Cohn et al., 1994; Settles, 2009). Our selective annotation step for in-context learning shares the same goal of reducing the annotation cost. Most active learning methods involve iterative parameter updates (e.g., Wang et al., 2017; Kasai et al., 2019), which are computationally expensive for large language models used in in-context learning. Similar to our vote-$k$ algorithm, Lin & Bilmes (2009) used the facility location objective to optimize representativeness. We observed that this objective largely underperforms vote-$k$ for in-context learning, probably due to the fact the vote-$k$ (1) is effectively tailored to the prompt retrieval step of in-context learning and (2) directly optimizes the diversity of selected samples (see §4.5). More recently, the effectiveness of active learning has been questioned when large-scale pretrained models are finetuned for various tasks (Karamcheti et al., 2021; D'Arcy & Downey, 2022). Our experiments (§3) showed that selective annotation helps reduce the annotation cost of in-context learning, departing from the recent observations on finetuning with active learning. We hypothesize that it is because in-context learning with similarity-based prompt retrieval is more robust to outliers since each test instance only retrieves its most similar examples. This is supported by §4.4, where *random* prompt retrieval does not benefit from selective annotation.

## 6 CONCLUSION

Much recent work illustrated the ability of large language models to adapt to new tasks simply from a few demonstration examples. We presented in-depth studies on the implications of this ability for dataset annotation through the lens of selective annotation and introduced an annotation-efficient practice. The best selective annotation method explored in this paper, our vote-$k$ method, selects diverse, representative examples to annotate. In terms of the task performance, vote-$k$ improves the performance on 10 diverse tasks by a large margin. Moreover, vote-$k$ selective annotation yields similar performance to state-of-the-art supervised finetuning with 10-100$\times$ less annotation cost. We further show that the effectiveness of vote-$k$ is consistent with different language model sizes and domain shifts between training and test data. We hope that our findings will help researchers and practitioners efficiently design new natural language tasks and beyond.

ACKNOWLEDGEMENTS

We thank Sewon Min, Pradeep Dasigi, Yanda Chen, Yushi Hu, Alisa Liu, and the ARK group at UW for their helpful feedback on this work.

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

| Dataset | Task | Examples |
|---------|------|----------|
| HellaSwag | Commonsense Reasoning | A woman is outside with a bucket and a dog. The dog is running around trying to avoid a bath. She...
  A) rinses the bucket off with soap and blow dry the dog's head.
  B) uses a hose to keep it from getting soapy.
✓C) gets the dog wet, then it runs away again.
  D) gets into a bath tub with the dog. |
| MRPC | Paraphrase Detection | Sales rose 37 per cent year-on-year to 1.76bn, beating expectations. Sales for the quarter beat expectations, rising 37 percent year-on-year to 1.76 billion euros.
→ ✓ Paraphrase |
| SST-5 | Sentiment Analysis | A warm, funny, engaging film.→Positive
Suffers from the lack of a compelling narrative.→Negative |
| MWoZ 2.4 | Dialogue State Tracking | I am looking for ALexender b&b
Dialogue state: **alexander bed and breakfast** |
| GeoQuery | Semantic Parsing | What is the area of California?
`SELECT` state.area `FROM` state `WHERE` state.
    state_name=`'california'` |
| DBpedia | Topic Classification | The keeled box turtle (Cuora mouhotii syn. Pyxidea mouhotii) is a species of turtle in the family Geoemydidae. It is native to Asia where it occurs in China India Laos Burma Vietnam Thailand and Bhutan. Other common names include keel-backed terrapin and jagged-shelled turtle.
Topic: **animal** |
| MNLI | Natural Language Inference | The F/A-18-E/F program eliminated over 40 percent of the parts used to build predecessor aircraft to make the design more robust for manufacturing and identified critical manufacturing processes, bringing them under control before the start of production. The new design with robustness also increased the safety of machines.
→ ✓ neutral |
| RTE | Natural Language Inference | Judie Vivian, chief executive at ProMedica, a medical service company that helps sustain the 2-year-old Vietnam Heart Institute in Ho Chi Minh City (formerly Saigon), said that so far about 1,500 children have received treatment. The previous name of Ho Chi Minh City was Saigon.
→ ✓ entailment |
| Natural Questions | Open-Domain QA | when was music first played on the radio
→ ✓ 1917 |
| XSUM | Summarization | Bliss said there was a shortage of neonatal nurses and doctors, and safety standards were not being met. ...... Dr Jenny Calvert, of the Wales Neonatal Network, said they are working to further develop medical training in neonatology to help recruit more trainee doctors.
Summary: **Neonatal services across Wales are overstretched and under pressure with the safety of vulnerable babies at risk, according to a charity.** |

Table 6: All of the 10 datasets with examples used in our experiments. The 10 datasets span various formations, including classification (SST-5, Socher et al., 2013; MRPC, Dolan et al., 2004), multiple-choice selection (HellaSwag, Zellers et al., 2019), and code/text generation (MWoZ 2.4, Budzianowski et al., 2018; GeoQuery, Zelle & Mooney, 1996; NQ, Kwiatkowski et al., 2019).

# Appendices

## A  DATASETS AND TASKS

## B  PROMPT TEMPLATES

### B.1  HELLASWAG

**Input:**

```
The topic is Grooming dog. Two women attempt to wash two dogs. they get
in the tub with the dogs and do shampoo, soap, and then rinse the dogs.
......
The topic is Bathing dog. A couple is outside with a bucket and a dog.
The dog is running around trying to avoid a bath. they
```

**Output:**

```
get the dog wet, then it runs away again.
```

### B.2  MRPC

**Input:**

```
Are the following two sentences 'equivalent' or 'not equivalent'?
This was around the time Congress was debating a resolution granting the
President broad authority to wage war ..
Within four days , the House and Senate overwhelmingly endorsed a
resolution granting the president authority to go to war ..
answer:not equivalent
......
Are the following two sentences 'equivalent' or 'not equivalent'?
Kerry last month outlined a U.N. resolution authorizing a military force
under U.S. command and transferring responsibility to the United Nations
for the political and humanitarian efforts ..
Kerry outlined last month a UN resolution authorizing a military force
under US command and transferring responsibility for political and
humanitarian efforts to the UN ..
answer:
```

**Output:**

```
equivalent
```

### B.3  SST5

**Input:**

```
How do you feel about the following sentence?
the movie 's blatant derivativeness is one reason it 's so lackluster .
answer:negative
......
How do you feel about the following sentence?
the movie 's something-borrowed construction feels less the product of
loving , well integrated homage and more like a mere excuse for the wan ,
thinly sketched story .
answer:
```

**Output:**

```
negative
```

### B.4 MULTIWOZ

**Input:**

```
CREATE TABLE hotel(
  name text,
  ......,
  internet text CHECK (internet IN (dontcare, yes, no))
)
/*
4 example rows:
SELECT * FROM hotel LIMIT 4;
name  pricerange  type  parking book_number_of_days book_day  book_people
area  stars internet
a and b guest house moderate guest house dontcare 3 friday 5 east 4 yes
......
/*
......
-- Using valid SQLite, answer the following multi-turn conversational
questions for the tables provided above.
Example #1
[context] hotel-area: west, hotel-stars: 3, hotel-internet: yes
[system] the hobsons house is available in that area .
Q: [user] that sounds like it will work . can i book that for 3 nights
starting wednesday ?
SQL: SELECT * FROM hotel WHERE book_day = wednesday AND book_people = 1
AND book_number_of_days = 3 AND name = hobsons house;
......
Example #22
[context] hotel-parking: yes, hotel-pricerange: moderate, hotel-type:
guest house, hotel-stars: 4
[system] there are 9 in the area . i recommend the warkworth house .
Q: [user] can you book that 1 for 4 nights starting on wednesday ?
SQL: SELECT * FROM
```

**Output:**

```
hotel WHERE book_day = wednesday AND book_number_of_days = 4 AND name =
warkworth house;
```

### B.5 GEOQUERY

**Input:**

```
CREATE TABLE "border_info" ("state_name" text, "border" text)
/*
state_name     border
   alabama tennessee
   alabama   georgia
   alabama    florida
*/
......
-- Using valid SQLite, answer the following questions for the tables
provided above.
-- which state has the longest river
SELECT RIVERalias0.TRAVERSE FROM RIVER AS RIVERalias0 WHERE RIVERalias0.
LENGTH = ( SELECT MAX( RIVERalias1.LENGTH ) FROM RIVER AS RIVERalias1 ) ;
......
-- what is the longest river in the state with the highest point
SELECT
```

**Output:**

```
RIVERalias0.RIVER_NAME FROM HIGHLOW AS HIGHLOWalias0 , RIVER AS
RIVERalias0 WHERE HIGHLOWalias0.HIGHEST_ELEVATION = ( SELECT MAX(
HIGHLOWalias1.HIGHEST_ELEVATION ) FROM HIGHLOW AS HIGHLOWalias1 ) AND
RIVERalias0.TRAVERSE = HIGHLOWalias0.STATE_NAME ORDER BY RIVERalias0.
LENGTH DESC LIMIT 1 ;
```

## B.6  DBPEDIA

**Input:**

```
title: Cupressus funebris; content:  Cupressus funebris (Chinese Weeping
Cypress) is a species of cypress native to southwestern and central
China. It may also occur naturally in Vietnam.
plant
......
title: Keeled box turtle; content:  The keeled box turtle (Cuora mouhotii
syn. Pyxidea mouhotii) is a species of turtle in the family Geoemydidae.
It is native to Asia where it occurs in China India Laos Burma Vietnam
Thailand and Bhutan. Other common names include keel-backed terrapin and
jagged-shelled turtle.
```

**Output:**

```
animal
```

## B.7  MNLI

**Input:**

```
Ideally, the design fixes for the failures should be corrected prior to
manufacturing production units.. Based on that information, is the claim
The fixes should be addressed before they reach the assembly line if this
was a smart plan. "True", "False", or "Inconclusive"?
answer:Inconclusive
......
The F/A-18-E/F program eliminated over 40 percent of the parts used to
build predecessor aircraft to make the design more robust for
manufacturing and identified critical manufacturing processes, bringing
them under control before the start of production.. Based on that
information, is the claim The new design with robustness also increased
the safety of machines.  "True", "False", or "Inconclusive"?
answer:
```

**Output:**

```
Inconclusive
```

## B.8  RTE

**Input:**

```
After giving nearly 5,000 people a second chance at life, doctors are
celebrating the 25th anniversary of Britian's first heart transplant
which was performed at Cambridgeshire's Papworth Hospital in 1979..\par
question: The first heart transplant in Britian was performed in 1979..
True or False?
answer:True
......
Judie Vivian, chief executive at ProMedica, a medical service company
that helps sustain the 2-year-old Vietnam Heart Institute in Ho Chi Minh
City (formerly Saigon), said that so far about 1,500 children have
received treatment..
question: The previous name of Ho Chi Minh City was Saigon.. True or
```

```
False?
answer:
```

**Output:**

```
True
```

## B.9  NATURAL QUESTION

**Input:**

```
Write an answer: who invented the radio during the industrial revolution
other
Guglielmo Marconi, 1st Marquis of Marconi
......
Write an answer: when was music first played on the radio
```

**Output:**

```
other
1917
```

## B.10  XSUM

**Input:**

```
Write a short summary
Health Minister Mark Drakeford said the money would be used to improve
areas of concern, including out-of-hours help and access to psychological
treatment.
......
money won't get the help they need in a timely fashion," she said.
TL;DR: An extra [Unicode token]7.6m a year will be invested to improve
mental health services for children and young people in Wales.
......
write a short summary:
Bliss said there was a shortage of neonatal nurses and doctors, and
safety standards were not being met.
......
Dr Jenny Calvert, of the Wales Neonatal Network, said they are working to
further develop medical training in neonatology to help recruit more
trainee doctors.
TL;DR:
```

**Output:**

```
Neonatal services across Wales are overstretched and under pressure with
the safety of vulnerable babies at risk, according to a charity.
```

## C    DETAILED MAIN RESULTS

This section provides a detailed version of our main results in Table 2, where the maximum performance and minimum performances among the three trials are reported. Results are shown in Table 7 and Table 8.

| Method | | Classification | | | | |
|---|---|---|---|---|---|---|
| $|\mathcal{L}|$ | Selection | MRPC | SST-5 | MNLI | DBpedia | RTE |
| 100 | Random | 63.5/66.0/60.5 | 44.2/47.3/41.8 | 37.4/41.0/33.2 | 89.8/91.0/88.3 | 51.5/53.9/48.4 |
| 100 | Vote-$k$ | **70.7**/72.3/69.1 | **53.0**/54.7/51.2 | **47.3**/50.0/44.5 | **93.4**/94.1/92.6 | **55.5**/57.0/53.9 |
| 18 | Random | 59.6/64.8/52.7 | 39.8/46.1/37.1 | 36.7/40.6/30.9 | 77.6/82.0/71.9 | 50.4/53.5/45.7 |
| 18 | Vote-$k$ | **64.2**/67.6/59.0 | **47.6**/50.0/44.5 | **41.0**/44.5/37.1 | **87.1**/90.6/85.2 | **54.3**/56.2/51.6 |

Table 7: Main result Table 2 with the mean/max/min results reported across three trials

| Method | | Multi-Choice | Dialogue | Generation | | |
|---|---|---|---|---|---|---|
| $|\mathcal{L}|$ | Selection | HSwag | MWoZ | GeoQ | NQ | XSum |
| 100 | Random | 65.2/66.4/63.3 | 47.2/49.2/44.5 | 78.6/80.5/77.3 | 30.8/32.8/28.1 | 15.3/16.4/14.8 |
| 100 | Vote-$k$ | **70.7**/71.5/69.5 | **51.4**/53.1/49.6 | **82.8**/83.6/82.0 | **33.6**/35.2/31.6 | **17.2**/17.6/16.4 |
| 18 | Random | 62.5/66.4/57.4 | 33.6/39.5/25.0 | 62.4/65.2/57.8 | 29.8/31.6/26.6 | 13.6/14.5/12.5 |
| 18 | Vote-$k$ | **67.4**/71.1/64.8 | **42.8**/47.7/40.2 | **72.5**/74.2/69.5 | **32.3**/33.6/30.1 | **15.2**/16.0/14.5 |

Table 8: Main result Table 2 with the mean/max/min results reported across three trials.

## D    EVALUATE HELLASWAG ON OPT-175B MODEL

Here we show that vote-$k$ also improves model performance for OPT-175B

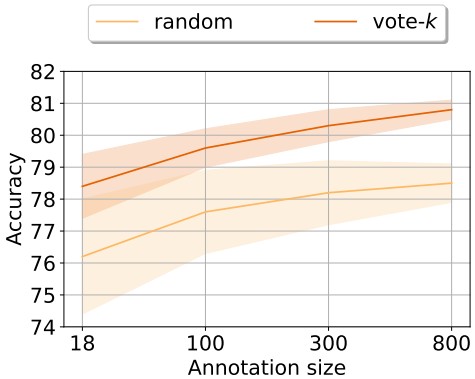

Figure 4: OPT-175B performance of ICL-random and ICL-vote-$k$ on HellaSwag

## E    REMOVING OUTLIERS FOR FINETUNING

Here we show that explicitly removing outliers also helps finetuning to benefit from vote-$k$.

## F    DIVERSITY AND REPRESENTATIVENESS OF SELECTED SAMPLES

We hypothesized that both representativeness and diversity are crucial for selective annotation (§2.1). Here we evaluate the diversity and representativeness of samples that are selected by different methods,

|  | Outliers not removed | | 10% outliers removed | |
|---|---|---|---|---|
|  | FT-random | FT-vote-$k$ | FT-random | FT-vote-$k$ |
| HellaSwag | 55.6 | 53.5 | 56.8 | 59.6 |
| MRPC | 56.3 | 55.6 | 57.9 | 60.4 |

Table 9: Effects of vote-$k$ in finetuning(FT) with annotation budget of 100. *10% outliers removed* means that we removed 10% of examples farthest to the training data, measured by average cosine similarity. The selection is conducted after example removal. After removing outliers, vote-$k$ selection improves the model few-shot performance.

using the methods from prior work on active learning (Margatina et al., 2021); their measures of diversity and representativeness use token overlap or embedding cosine similarities. As shown in Table 10, K-means improves both the diversity and the representativeness, as compared to random selection, while vote-$k$ further improves it.

| Method | DIV-I | | | DIV-F | | | REPR. | | |
|---|---|---|---|---|---|---|---|---|---|
| Selection | HellaSwag | SST-5 | MWoZ | HellaSwag | SST-5 | MWoZ | HellaSwag | SST-5 | MWoZ |
| Random | $0.182_{0.007}$ | $0.099_{0.003}$ | $0.368_{0.008}$ | $0.415_{0.008}$ | $0.317_{0.004}$ | $0.675_{0.006}$ | $0.558_{0.007}$ | $0.424_{0.003}$ | $0.696_{0.004}$ |
| K-means | $0.184_{0.006}$ | $0.102_{0.002}$ | $0.372_{0.003}$ | $0.417_{0.006}$ | $0.317_{0.002}$ | $0.676_{0.004}$ | $0.559_{0.005}$ | $0.423_{0.002}$ | $0.698_{0.003}$ |
| Vote-$k$ | 0.191 | 0.108 | 0.379 | 0.425 | 0.321 | 0.683 | 0.565 | 0.426 | 0.702 |

Table 10: DIV-I refers to diversity in input space, which measures the diversity of selected data in the input feature space, i.e., raw text; DIV-F refers to diversity in feature space, which measures the diversity in the dense feature space, i.e., sentence embeddings; REPR. refers to representativeness, which measures the representativeness of selected data. Subscripts stand for standard deviation.

## G    DETAILS OF SELECTIVE ANNOTATION METHODS

In this section, we provide details of selective annotation methods used in Section 4.5.

### G.1    VOTE-$k$ SELECTIVE ANNOTATION

Algorithm 1 describes the vote-$k$ selective annotation method introduced in Section 2.1.

### G.2    GREEDY ALGORITHM FOR MAXIMIZING FACILITY LOCATION

Lin & Bilmes (2009) proposed to maximize the facility location objective to optimize representativeness of the selected samples. Since this objective satisfies the submodular property, they applied a greedy algorithm as an approximation. Algorithm 2 describes the selective annotation method adapted from this greedy algorithm.

### G.3    EMBEDDING DIVERSITY

This method aims to find diverse samples to annotate using embedding vectors. We first compute a vector representation for each *unlabeled* training instance by Sentence-BERT (Reimers & Gurevych, 2019), which is a variant of BERT (Devlin et al., 2019), finetuned to detect paraphrases.[6]  For instance, consider an example from SST-5 sentiment analysis in Table 6:*A very well-made, funny and entertaining picture*. We simply run Sentence-BERT on this text input and average the resulting vectors over the words to obtain a vector representation.

Once embeddings are computed for all training data, we use them to find a diverse set of training instances. The intuition here is that a diverse set of annotated examples facilitates the subsequent prompt retrieval step since similar in-context examples can be found for many test instances. To find

---

[6]https://huggingface.co/sentence-transformers/all-mpnet-base-v2.

---

**Algorithm 1** Voke-k Selective Annotation

---

1: **Input:** $\mathcal{X} = \{x_i\}_{i=1}^N$: a set of unlabeled samples; $M$: the number of samples to be selected; LM: inference language model.
2: **Initialization:** $\mathcal{L} = \varnothing, \mathcal{U} = \mathcal{X}$. $G = (V, E)$, where $V = \mathcal{X}$ and $(u, v) \in E$ if $v$ is one of $u$'s $k$ nearest vertices in terms of the cosine similarity between the embeddings.
3: **while** $|\mathcal{L}| < M/10$ **do**
4:   $u^* = \arg\max_{u \in \mathcal{U}} \sum_{v \in \{v|(v,u) \in E, v \in \mathcal{U}\}} s(v)$, where $s(v) = \rho^{-|\{\ell \in \mathcal{L}|(v,\ell) \in E\}|}$, $\rho > 1$
5:   $\mathcal{L} = \mathcal{L} \cup \{u^*\}$
6:   $\mathcal{U} = \mathcal{U} \setminus \{u^*\}$
7: **end while**
8: **for** $u$ in $\mathcal{U}$ **do**        ▷ Compute the confidence score of each instance.
9:   $\text{score}(u) = \frac{1}{\mathbf{q}} \sum_t \log p(q_t|\mathbf{q}_{<t}, \mathbf{z}; \Theta)$, where $p$ is LM prediction function and $\Theta$ is LM parameters
10: **end for**
11: **for** $j = 1, \ldots, 9/10M$ **do**
12:   $\mathcal{U}_j = \text{indices}[(j-1)|\mathcal{U}|/M : j|\mathcal{U}|/M]$        ▷ Divide the examples by confidence score.
13:   $u^* = \arg\max_{u \in \mathcal{U}_j} \sum_{v \in \{v|(v,u) \in E, v \in \mathcal{U}_j\}} s(v)$, where $s(v) = \rho^{-|\{\ell \in \mathcal{L}|(v,\ell) \in E\}|}$, $\rho > 1$
14:   $\mathcal{L} = \mathcal{L} \cup \{u^*\}$
15: **end for**
16: **Return:** $\mathcal{L}$: selected samples.

---

**Algorithm 2** Greedy Algorithm for Facility Location Objective

---

1: **Input:** $\mathcal{U} = \{x_i\}_{i=1}^N$: a set of unlabeled samples; $M$: the number of samples to be selected.
2: **Initialization:** $\mathcal{L} = \varnothing, \mathcal{U} = V$. $\forall i, \rho_i = -1$: maximum similarity of $x_i$ to selected samples.
3: **while** $|\mathcal{L}| < M$ **do**
4:   $u^* = \arg\max_{u \in \mathcal{U}} \sum_{i=1}^N (\max\{0, \cos(x_i, x_u) - \rho_i\})$
5:   $\mathcal{L} = \mathcal{L} \cup \{u^*\}$
6:   $\mathcal{U} = \mathcal{U} \setminus \{u^*\}$
7:   $\forall i, \rho_i = \max\{\rho_i, \cos(x_i, x_{u^*})\}$    // update maximum similarity of each $x_i$ to selected samples
8: **end while**
9: **Return:** $\mathcal{L}$: selected samples.

---

a set of diverse embeddings, we take a simple, iterative approach: in every iteration, we choose an instance furthest from the already chosen ones. Specifically, let $\mathcal{L}$ and $\mathcal{U}$ denote the sets of already chosen (i.e., labeled) samples and unlabeled samples, respectively. Suppose also that $M$ is the target number of labeled examples (i.e., the annotation budget). Then, in every iteration, we choose the unlabeled sample that has the largest total cosine distance from $\mathcal{L}$: $\arg\min_{u \in \mathcal{U}} \sum_{\ell \in \mathcal{L}} \cos(u, \ell)$. Here we abuse $u$ and $\ell$ to mean both the instances and their embedding vectors from Sentence-BERT. The first labeled sample is randomly selected from the 3K unlabeled examples (§3.1), and the iterative process continues until $|\mathcal{L}| = M$.

## H  LABEL DISTRIBUTION IN SELECTIVE ANNOTATION

We calculate the ratio between the numbers of the most frequent class and the least frequent class in the selected 100 instances. For example, 53.3 in the Ori. (original dataset) column indicates that the original dataset contains 53.3 times more examples with the most frequent label than those with the least frequent label. As shown in Table 11, with random selection, the ratio is similar to the original dataset. With selective annotation, the imbalance problem is significantly alleviated.

## I  T-SNE VISUALIZATION OF SELECTIVE ANNOTATION

We compare examples from selective annotation and full training data using the t-SNE visualization van der Maaten & Hinton (2008). As shown in Figure 5, vote-$k$ selects diverse (better coverage) and representative (excluding outliers) instances from the task space across datasets.

|  | Ori. | Random | MFL | K-means | Diversity | Least-conf | Conf-only | Fast vote-$k$ | Vote-$k$ |
|---|---|---|---|---|---|---|---|---|---|
| Amazon | 53.3 | 59.0 | 55.0 | 52.0 | 20.5 | 48.0 | 26.5 | 14.0 | **11.0** |
| CivilComments | 7.8 | 11.5 | 9.0 | 7.3 | 5.7 | 15.7 | 6.7 | 5.3 | **4.6** |

Table 11: The ratio between the number of examples with the most frequent class and those with the least frequent class, when different selective annotation methods are applied to choose 100 instances. Ori. denotes the ratio in the original dataset. vote-$k$ significantly reduces the ratio and alleviates the label imbalance problem.

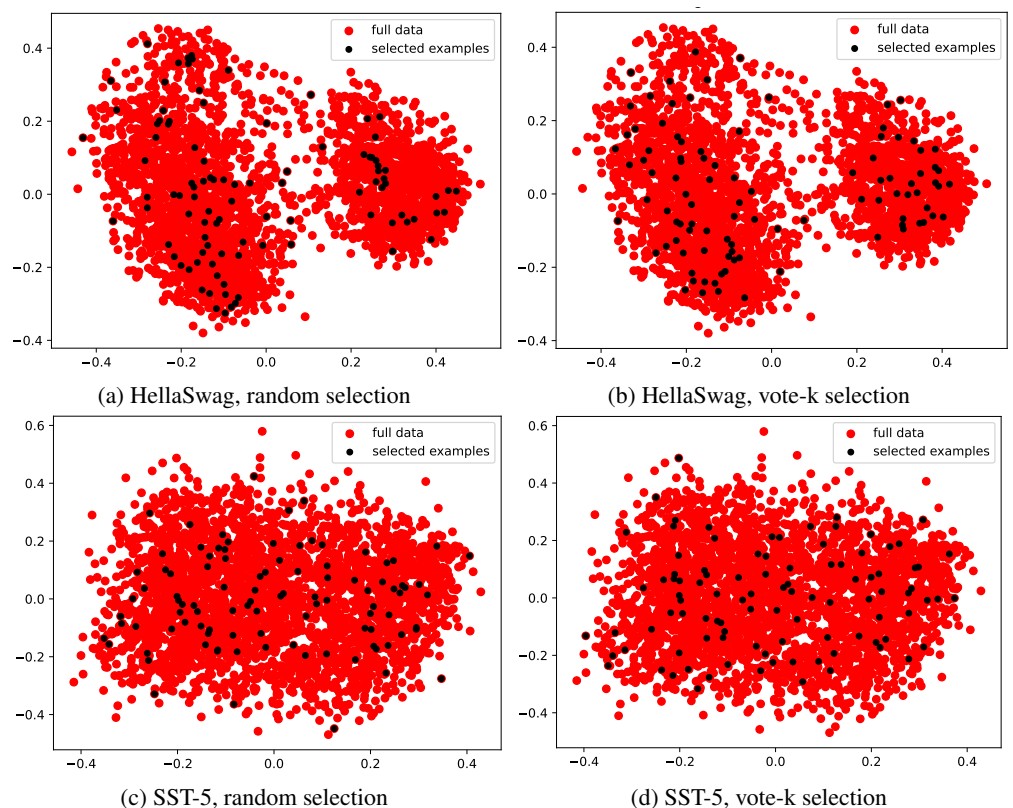

(a) HellaSwag, random selection

(b) HellaSwag, vote-k selection

(c) SST-5, random selection

(d) SST-5, vote-k selection

Figure 5: t-SNE visualization of the 100 randomly/vote-$k$ selected examples in the dataset space. vote-$k$ selects more diverse and representative examples, compared to random selection.

## J SIMILARITY BETWEEN CONTEXT EXAMPLES AND INPUT

We calculate the average cosine similarity between the context examples and the input text in three representative datasets, HellaSwag, SST-5 and MWoZ. As shown in the Table 12, without similarity-based retrieval in the second step, the context examples and input text are not very similar when the first step is vote-$k$ selection. We suspect the reason is that vote-$k$ selects diverse instances and random retrieval includes irrelevant examples in the prompt. **This explains the poor performance of vote-$k$ with random retrieval in Table 4**.

## K SELECTED EXAMPLES

In Table 1314, we provide a few examples from random selection and vote-$k$ selection, when the annotation size is 18.

| Method | | | Dataset | | |
|---|---|---|---|---|---|
| $|\mathcal{L}|$ | Selection | Retrieval | HellaSwag | SST-5 | MWoZ |
| 100 | Vote-$k$ | Similar | 0.3393 | 0.1747 | 0.4881 |
| 100 | Random | Similar | 0.3134 | 0.1535 | 0.2212 |
| 100 | Vote-$k$ | Random | 0.0057 | 0.0570 | 0.0006 |
| 100 | Random | Random | 0.0180 | 0.0065 | 0.0016 |

Table 12: Average cosine similarity scores between the context examples and evaluation input. With similarity retrieval in the second step, vote-$k$ selection always helps to find more similar examples in the prompt.

| Dataset | Randomly selected examples | Vote-$k$ selected examples |
|---|---|---|
| SST-5 | simple , poignant and leavened with humor , it 's a film that affirms the nourishing aspects of love and companionship .→Positive
this is a nicely handled affair , a film about human darkness but etched with a light -lrb- yet unsentimental -rrb- touch .→Positive
what is the filmmakers ' point ?→Neutral
what really happened ?→Neutral
an admirable , sometimes exceptional film.→Very Positive | take away the controversy , and it 's not much more watchable than a mexican soap opera .→Very Negative
any rock pile will do for a set .→Negative
a visual spectacle full of stunning images and effects .→Very Positive
the most consistently funny of the austin powers films .→Positive
occasionally funny , sometimes inspiring , often boring .→Neutral |

Table 13: Demo examples in SST-5 from random selection and vote-$k$ selection. In general, vote-$k$ selects more diverse and representative instances, with a more balanced label distribution.

| Dataset | Randomly selected examples | Vote-$k$ selected examples |
|---|---|---|
| HellaSwag | A man is completing a rubiks cube. a timer. . .
  A) begins showing the amount of time it will take it to complete the disk.
  B) is counting down on the side of the cube.
  ✓C) is sitting on the table next to him.
  D) goes hand at the end of the board while the man continues to solve and figure out the cubes.
A man is completing a rubiks cube. A timer is sitting on the table next to him. he. . .
  A) takes the cube and couches it for 10 seconds.
  B) finishes the rubiks cube and sets it aside.
  C) starts hurting himself trying to solve the rubiks cube.
  ✓D) sets the completed rubiks cube down on the table.
A client tap with the finger inside a square. Then, the person stacks all the cards and puts the tokens inside the squares. then people. . .
  ✓A) start to gamble, while the woman distribute the cards and pick up tokens.
  B) play hole dig while other people watches.
  C) pass a disk on the tile tile.
  D) five card get tokens inside the square.
After, the client shows with the hand, then the person make gestures with the hand showing the table. next the woman. . .
  ✓A) pick up the cards.
  B) change the angle of the arm and shows her knee cucumbers.
  C) puts brushes on sides of the table and open the places one by one with the hand giving a thumbs up and different gestures to the client.
  D) place the blindfold on the table, then she lay on her back.
He takes a scour and begins to scour the wall paper. He uses a paint roller to soak the wall paper for easy removal. he. . .
  A) points to several things on the wall and sprays the wall paper.
  B) takes scissors and measures the wall paper closely.
  ✓C) then demonstrates how steam can be used too to loosen the wall paper.
  D) wipes the wall paper down against the wall. | Mj's mommy is playing around with her hair smoothing it out. She goes to the bathroom with clips in her hair and gets out treatments and shampoos. she. . .
  A) laughs until she is full of tears.
  B) puts a little blow dryer on her hair and turns it on over and over.
  ✓C) begins to apply the hair things to her hair and combing it out, she starts to part her hair and put in curlers.
  D) gets in the bathroom dynain her hair with an iron in her shirt.
The batter is fighing with the man and other people trye to calm him down. a lo of people wearing black unifroms. . .
  ✓A) are running in the field fighting.
  B) are all sitting on the floor.
  C) are playing cornstarch ball against each other on the field.
  D) are cools the batter.
A group of athletes row on canoes during a race in between buoys on a waterway. the men. . .
  A) pass over a wooden structure in the river.
  B) paddle while crashing through endless waves in the river.
  ✓C) cross the final numbered buoys and glide while slowing down after the race.
  D) go over large cliffs into a lagoon.
A pair of handlebars are detached from the bike and is laying down flat on a table. the man. . .
  A) unstraps the handlebars and puts them back up and begins to pedal of his bike.
  B) rolls them up and ties the handles with a pulley.
  C) then shows with his knife and begins to a process of cutting them in half and putting them back to the bike.
  ✓D) then picks up the handlebars and sticks them back into the bike tightening them with a key.
A man is completing a rubiks cube. a timer. . .
  A) begins showing the amount of time it will take it to complete the disk.
  B) is counting down on the side of the cube.
  ✓C) is sitting on the table next to him.
  D) goes hand at the end of the board while the man continues to solve and figure out the cubes. |

Table 14: Demo examples in HellaSwag from random and vote-$k$ selection. In general, vote-$k$ selects more diverse and representative instances

