# OpenReview forum: "Selective Annotation Makes Language Models Better Few-Shot Learners"
_ICLR.cc/2023/Conference — ICLR 2023 poster_

### Official Review · Reviewer_dEqn · 2022-10-22

**Confidence:** 3
**Correctness:** 3
**Technical Novelty And Significance:** 2
**Empirical Novelty And Significance:** 2
**Recommendation:** 5

**Clarity, Quality, Novelty And Reproducibility:**

The paper is well-written.
It is easy for others to reproduce the method.
I do not think the proposed method is novel, though it is effective. It is an ensemble of existing techniques.

**Strength And Weaknesses:**

Strength:
The proposed method is simple yet effective.
The paper is well-written and thus easy to follow.
The empirical studies provide interesting insights about how the method works.

Weaknesses:
Some important alternatives and baselines are not involved in the comparison.
There is inconsistency in presentation.

**Summary Of The Paper:**

The paper proposes an active learning approach to select annotated examples for in-context learning in large PLMs. The proposed method is made up of three components: (1) nearest neighbors: diverse examples are selected from a graph constructed with nearest neighbors; (2) pre-trained language model: more examples are selected according to the scores of the PLM; and (3) prompt retrieval. Empirical studies on a number of public datasets demonstrate the effectiveness of the proposed method in comparison with random selection.

**Summary Of The Review:**

The paper proposes an active learning approach for in-context learning for large PLMs.

The method is simple yet effective, and the insights from the experiments are interesting. However, I have a few concerns:
Regarding to comparison:
(1) can you compare vote-k with the following alternative? (a) replace vote-k with random selection; (b) run PLM and obtain confident scores; and (c) select annotations according to the confident scores.
(2) can you compare the proposed method with some soft prompt tuning method?

Regarding to presentation.
From the description in Page 3, it seems that the remaining 9M/10 examples are selected according to the confident scores, but  Line 11-16 in Algorithm 1 has nothing to do with the confident scores.

---

> ### Author Response · Authors · 2022-11-14
> **Thank you for your review**
>
> Thank you for your review and suggestions to improve the clarity of the paper. Below we address the concerns and questions raised in the review. We are happy to hear that the reviewer found our method simple and effective. We used purple text to clarify the parts that we modified in the revision.
>
> **Baselines** We added the suggested baseline using a pretrained language model, where (a) replace vote-k with random selection; (b) run PLM and obtain confident scores; and (c) select annotations according to the confident scores. See our Table 5 (also shown below) in the revision (denoted as Conf-only). It outperforms several other methods but still underperforms both fast vote-*k* and vote-*k*.
>
> |                    | Random |    MFL   | K-means | Diversity | Least-conf | Conf-only    |  Fast vote-*k* | Vote-*k* |
> | -----------    | ----------|  ---------| -----------| ----------| -------------| -------------- | ----------------| ----------|
> | HellaSwag  |   65.2     |     66.5   |    67.6     |  68.2     |        68.4     |        68.6      |          69.5       |    70.7    |
> | SST-5          |   44.2    |     45.6   |    47.2     |  48.5     |        46.2     |        48.3      |          51.9       |    53.0     |
> | MWoZ         |   47.2    |     48.3   |    48.8     |  49.2     |        49.4     |        49.2      |          50.2       |    51.4     |
>
> Regarding the comparison to the soft prompt, we want to highlight that 1) our paper focuses on selective annotation for reducing labeling costs as opposed to finding/tuning better prompts; 2) it is computationally (and financially) expensive to even just tune prompts with large language models with 175B parameters used in our experiments.
>
> Finally, we also note that we added the k-means baseline, following Reviewer ZPh1’s suggestion.
>
> **Novelty**
> The reviewer raises a concern about the novelty of the proposed method. As the reviewer ZPh1 points out, the proposed use-case in this work is interesting and novel. Constructing graphs of unlabeled examples for sample selection is not the main novelty in this work, but novelty here is (1) in combining these ideas to make a practical system, and (2) the experimental work demonstrating insights comparing this with fine-tuning and other selective annotation schemes. The main focus in this paper is NOT to propose a new example selection method (although we indeed find a simple yet effective method to select examples and save annotation costs), instead, we aim to propose a new paradigm that is practically useful, e.g., when a new task comes, using in-context learning with selective annotation helps to achieve the best performance with the least manual labeling. In the fine-tuning, active learning literature showed that example selection is not very effective for pretrained large language models (discussed in the related work section), while we demonstrated that selective annotation can significantly enhance the model performance under in-context learning.
>
> **Confusion regarding the vote-k algorithm** The remaining 9M/10 examples are selected according to the confidence scores and voting. The confidence scores are computed in Line 8 of Algorithm 1 in Appendix G. Then in Line 12, the examples are divided according to the confidence scores. We clarified this point in the revised version (see our algorithm comments in the revision).

---

> ### Author Response · Authors · 2022-12-04
> **Follow-up with reviewer**
>
> Thanks for your valuable comments. We are writing this follow-up post because it has been days since we posted our initial response. Please let us know if your initial concerns are addressed, and we would appreciate any further questions or comments.

---

### Official Review · Reviewer_bT5a · 2022-10-25

**Confidence:** 4
**Correctness:** 2
**Technical Novelty And Significance:** 2
**Empirical Novelty And Significance:** 2
**Recommendation:** 5

**Clarity, Quality, Novelty And Reproducibility:**

Apart from some of my comments above, the rest of the paper is well-written and easy to understand.

**Strength And Weaknesses:**

### Strength

The efficient approach to select a pool of examples to annotate from unlabeled data which are shown to be diverse. The authors evaluate their method on several tasks including classification, reasoning, dialogue, semantic parsing and generation.

The authors also provide a detailed analysis of their method compared to other baselines.

### Weaknesses
I have a few comments:

* Do the authors take the cost of encoding every unlabeled training instance using Sentence-BERT into account when comparing methods? I believe this was not mentioned in the paper.
* In section 2.1, It is not very clear how the first M/10 examples are labeled. It is mentioned that “the current labeled $\mathcal{L}$ has M/10 samples” which is used to label other instances. Are they manually labeled?
* It is stated that the vote-k method encourages diversity. Since all the labels are removed first, does this method encourage or keep inclusivity of all the classes? Have the authors looked at whether their method favors examples from certain classes?
* In section 2.1, “we conduct experiments three times and report the average score”. What/where is the randomness? Examples selected or model prediction?
* Figure 2 shows a direct comparison between fine-tuned RoBERTa large and models such as DaVinci-002 while it is stated that the authors do not aim to conduct a head-to-head comparison. What is the comparison the authors are making here and perhaps could it be shown better with another figure? Results from this direct and not fair comparison are also mentioned in the abstract which could be misleading.
* In section 4.4, I think some more detail is needed as to why the combination of vote-k and similarity retrieval works, but vote-k alone with random selection of supporting examples has almost the same score as random selection.



**Summary Of The Paper:**

This paper proposes a method to select a pool of diverse examples to be annotated before evaluation with in-context learning. During evaluation, a cosine-similarity based method is used to retrieve supporting examples from this pool. The combination of these methods outperforms randomly selecting examples.

Key contribution of the paper is the selective annotation method called vote-k to make a pool of diverse examples.


**Summary Of The Review:**

The proposed method to collect a pool of examples and annotate them performs well in combination with similarity based retrieval for support examples for in-context learning. The authors motivated this approach well, and a detailed analysis is carried out. Some parts about the proposed method are not clear, and there are a few missing and potentially misleading pieces in the paper.

---

> ### Author Response · Authors · 2022-11-14
> **Thank you for your review**
>
> Thank you for your detailed review and important questions. We are glad to hear that the reviewer found the proposed paradigm efficient and effective, the evaluation is comprehensive and the analysis is detailed. Below we address the concerns and questions raised in the review. We used purple text to clarify the parts that we modified in the revision.
>
> **Computational cost for encoding** As the reviewer points out, we did not mention the cost of encoding queries in the paper. This cost is relatively small; for example, it only takes Sentence-BERT 7 minutes to encode 560,000 sentences using one A100 GPU with 40GB memory on the DBpedia dataset. We also highlight that this paper focuses on reducing human annotations, while keeping computational costs reasonably low.
>
> **How selected examples are annotated** In vote-k, the first M/10 examples are annotated by humans, which serve as demonstrations to get confidence scores used in further selection of 9*M/10 examples. The 9*M/10 examples are then annotated by humans, resulting in a total of M examples with human labels.
>
> **Label distribution** The selective annotation is based entirely on similarities derived from sentence embeddings, not label distributions. In practice, we often do not know the label distribution on the evaluation/deployment data. For example, the label distribution changes drastically when there is domain shift (see our experiments on the WILDS benchmark in Section 4.4). We thus believe that selective annotation should not be performed based on label distribution. We added an additional analysis in Appendix H, where we calculate the imbalance ratio (the ratio between the most frequent class to the least frequent class) from different selection methods. As seen in Table 11 (also shown below), vote-k indeed results in more balanced label distributions (lower imbalance ratio) despite the fact that vote-k is performed based on input representations. See also our response to Reviewers ZPh1 and 3Ufu.
>
> |                        |       Ori.     |  Random  |     MFL     | K-means  |  Diversity |Least-conf|Conf-only |Fast Vote-k|   Vote-k   |
> | ---------------- | ----------- | ----------- |-----------  | ----------- | ----------- |----------- |-----------  | ------------ | ----------- |
> |      Amazon     |      53.3     |    59.0      |      55.0     |    52.0      |     20.5      |     48.0    |     26.5     |      14.0      |   11.0       |
> |CivilComments|      7.8      |    11.5      |       9.0      |      7.3      |       5.7      |     15.7     |      6.7     |        5.3      |     4.6        |
>
> **Randomness is only for random selection** As discussed in Section 2.1, we conducted experiments three times and report the average score for **the random selection baseline**. Given a fixed unlabeled dataset, vote-k is indeed deterministic.
>
> **Head-to-head comparison** We did not conduct head-to-head comparisons because it is infeasible to finetune large models such as GPT DaVinci-002 in most cases. The comparison here illustrates that with the same number of labeled examples, large language models with in-context learning and selective annotation outperform smaller language models with finetuning, implying that large language models with in-context learning under selective annotation are promising. To clarify, we modified the figure legend of Figure 2 (now it explicitly mentions which pretrained model is used).
>
> **Novelty**
> As the review ZPh1 points out, we not only propose a sample yet effective selective annotation method, but more novelty includes combination of constructing graphs for unlabeled examples and similarity-based prompt retrieval to make a practical system. In addition, the experimental work demonstrating insights comparing this with fine-tuning and other selective annotation schemes.
>
> **Vote-k with random retrieval** As the reviewer points out, vote-k selection + random retrieval yields performance similar or worse compared to random selection + random retrieval (the bottom two rows in Table 4). We suspect that this is because if random retrieval is used on top of vote-k, it results in in-context examples **dissimilar** to input instances due to vote-k’s selection diversity. It is thus necessary to use similarity-based retrieval to benefit from vote-k. In the revision, we added a table that shows average similarity scores between in-context examples and test input instances (Table 12, also shown below). The results indeed suggest that vote-k + random retrieval suffers from the dissimilarity between in-context examples and test instances.
>
> |       |**Method**|      |       |**Dataset**|      |
> | --- | ----- | ---- |----  | --- | --- |
> |*L*  |  Selection | Retrieval  |HellaSwag| SST-5 | MWoZ  |
> |100 |Vote-*k*|Similar|0.3393| 0.1747| 0.4881 |
> |100 |Random|Similar|0.3134| 0.1535| 0.2212 |
> |100 |Vote-*k*|Random|0.0057| 0.0570| 0.0006 |
> |100 |Random|Random|0.0180| 0.0065| 0.0016 |

---

> ### Author Response · Authors · 2022-12-04
> **Follow-up with reviewer**
>
> Thanks for your valuable comments. We are writing this follow-up post because it has been days since we posted our initial response. Please let us know if your initial concerns are addressed, and we would appreciate any further questions or comments.

---

### Official Review · Reviewer_3Ufu · 2022-10-27

**Confidence:** 3
**Correctness:** 4
**Technical Novelty And Significance:** 3
**Empirical Novelty And Significance:** 3
**Recommendation:** 6

**Clarity, Quality, Novelty And Reproducibility:**

The paper is well-written. The proposed technique has been empirically validated well and code for this work has been made available to encourage reproducibility.

**Strength And Weaknesses:**

Strengths:
1. The authors present a simple technique for in-context learning with large language models that achieves consistently good results across a variety of NLP tasks.
2. The proposed ideas in isolation do not yield significant improvements. The authors find that the main ingredients for the success of in-context learning are a combination of selective annotation with similarity-based prompt retrieval.
3. The experimental section is thorough. Along with ablations and trying LMs of varying sizes, their technique is compared against many other existing selective annotation approaches and shown to consistently outperform the latter.

Weaknesses: Some of the design choices need to be elaborated on further and additional analysis-based experiments would also be useful. I elaborate on these further below for the authors to address.

- For the classification tasks, what do the label distributions look like over the labeled subset $\mathcal L$? Since the selective annotation is based entirely on similarities derived from sentence embeddings, there is nothing explicit ensuring that the label distribution over the selected subset is not skewed. Or, is the label distribution on the annotated subsets derived via vote-*k* indeed skewed for some tasks and the performance improvements are mainly coming from improvements on the labels that are well-represented? Some discussion of this would be useful.
- It seems like the optimal value of $k$ in vote-*k* would depend on the number of instances in the unlabeled set that changes with the tasks. A single value of $k=150$ was chosen for experiments across all tasks. Could the authors justify this choice further?
- From the diversity and representativeness measures shown in Table 10 in Appendix F, the difference between Random and vote-*k* does not appear to be very large. In the least, it's unclear how to assess the differences shown in this table. A qualitative analysis might be more revealing here. It might be interesting to see some examples, especially when the annotation budget is 18, of the kinds of instances that get selected depending on the task. t-SNE plots of these selected examples in the larger context of unlabeled instances might also be a good visualization to show. Please comment.

**Summary Of The Paper:**

In this work, the authors propose a two-step technique for improved few-shot learning with large language models on a variety of NLP tasks:
1. *Selective annotation*, an unsupervised similarity-based procedure referred to as vote-*k*, that selects diverse and representative examples that are subsequently labeled, and
2. *Prompt retrieval* at test time that retrieves in-context examples from the labeled set that are most similar to each test instance.

vote-*k* is found to improve performance on 10 diverse tasks by a large margin compared to random selections. Under similar budget constraints, in-context learning is shown to be a much better few-shot learner compared to standard model fine-tuning. vote-*k* is also shown to outperform existing methods for selective annotation.

**Summary Of The Review:**

I have rated this submission as a marginal accept for now. Look forward to the authors' responses to my questions.

---

> ### Author Response · Authors · 2022-11-14
> **Thank you for your review**
>
> Thank you for your detailed review and suggestions to improve the paper. We are glad to hear that you found that our experiments are comprehensive and our results are strong. Below we address the concerns and questions raised in the review. We used purple text to clarify the parts that we modified in the revision.
>
> **Label distribution** We agree that the selective annotation is based entirely on similarities derived from sentence embeddings, not label distributions. In practice, we often do not know the label distribution on the evaluation/deployment data. For example, the label distribution changes drastically when there are domain shifts (see our experiments on the WILDS benchmark in Section 4.4). We thus believe that selective annotation should not be performed based on label distribution.
>
> Although the selective annotation is unsupervised, our vote-k selective annotation explicitly emphasizes the diversity and representativeness of the selected subset, which is well supported by the quantitative measures in Table 10. Intuitively, a skewed subset with most instances coming from the same class is less likely to be diverse, and diversity implicitly enhances more balanced label distribution in the selected subset. In Appendix H, we added an additional analysis, where we calculate the label imbalance ratio (the ratio between the most frequent class to the least frequent class) for different selection methods. As seen in Table 11 (also shown below), vote-k indeed results in more balanced label distributions (lower imbalance ratio) despite the fact that vote-k is performed based on input representations. See also our response to Reviewer ZPh1.
>
> |                        |       Ori.     |  Random  |     MFL     | K-means  |  Diversity |Least-conf|Conf-only |Fast Vote-k|   Vote-k   |
> | ---------------- | ----------- | ----------- |-----------  | ----------- | ----------- |----------- |-----------  | ------------ | ----------- |
> |      Amazon     |      53.3     |    59.0      |      55.0     |    52.0      |     20.5      |     48.0    |     26.5     |      14.0      |   11.0       |
> |CivilComments|      7.8      |    11.5      |       9.0      |      7.3      |       5.7      |     15.7     |      6.7     |        5.3      |     4.6        |
> *Ori. denotes the ratio in the original dataset*
>
> **Choice of k** We select k to encourage the diversity and representativeness of selected examples. If k is too large, the graph will result in many edges and become similar to random selection. If k is too small, a small number of edges will be overrepresented, resulting in selection of outliers. In our experiments, we tried k=100, 150, 200 and found they led to no substantial performance differences. In practice, this hyper-parameter can be fixed for easy adaptation to many tasks, as in our experiments. We clarified this point in the revision.
>
> **Diversity and representativeness measure** We agree that the diversity numbers do not look very different in Table 10. This is due to the nature of similarity scores from [Margatina et al.](https://arxiv.org/pdf/2109.03764.pdf) (e.g., see their Table 3). Following the reviewer’s suggestion, we added some examples that compare random selection and selective annotation in Appendix K. Again following the reviewer’s suggestion, we added t-SNE plots in the revision (Appendix I).

---

> ### Author Response · Authors · 2022-12-04
> **Follow-up with reviewer**
>
> Thanks for your valuable comments. We are writing this follow-up post because it has been days since we posted our initial response. Please let us know if your initial concerns are addressed, and we would appreciate any further questions or comments.

---

### Official Review · Reviewer_ZPh1 · 2022-11-02

**Confidence:** 3
**Correctness:** 3
**Technical Novelty And Significance:** 3
**Empirical Novelty And Significance:** 3
**Recommendation:** 8

**Clarity, Quality, Novelty And Reproducibility:**

### Clarity:
I think the paper is well written overall, with a clear presentation. Aside from a small question about algorithm initialization, I felt like I understood the work.

### Quality:
Both the proposed algorithm and experimental evaluations presented seem high quality and interesting.

### Novelty:
The novelty of this work is mixed: the idea of using retrieval for in-context learning prompts is not new (this work cites Liu et al., 2022 and Rubin et al., 2022).. The idea of constructing graphs of unlabeled examples is also not new (common in label propagation literature). But novelty here is (1) in combining these ideas to make a practical system, and (2) the experimental work demonstrating some insights comparing this with fine-tuning and other selective annotation schemes.

### Reproducibility:
Given that the code for vote-k be released, there should be no impediments to reproducing the results presented in the paper.


**Strength And Weaknesses:**

## Strengths

1. In-context learning is a hot, rapidly evolving area of research and of high interest to the ICLR and NLP communities.  This paper presents a thorough set of evaluations of the impact of the choice of examples to include in the prompt, demonstrating that this can have a huge impact in final model behavior.

1. The paper is clear and easy to read. The evaluation section contains several interesting results that should be valuable to share with the broader community.

1. The gains of vote-k over other existing active learning methods presented is quite strong (Section 4.5).

1. The system proposed seems to be practical and relatively easy to implement in practice. As such, there is a chance that it could be applied in practice in real world applications of in-context learning.

1. The idea of automatically constructing graphs of unlabeled examples using similarity measures is relatively old in the label propagation literature. The proposed use-case in this work is interesting and novel.

## Weaknesses

1. The main weakness of the paper is the lack of a thorough description and analysis (including ablations) of the vote-k algorithm and its relationship to existing clustering algorithms. I would have hoped for a more complete discussion in Section 2.1, including the contents of Appendix G which seem important for readers to fully understand the proposed vote-k algorithm.

1. One particularly confusing part of vote-k: it was not immediately clear to me how this differs or is better than simple k-means clustering to discover centroids? Once a weighted graph is constructed (as described in Section 2.1, using cosine similarities of sentence-level representations), it should be possible to run any graph-based algorithm for finding centroids. I think a naive baseline (that does not use model prediction confidence scores) would be to use k-means.

1. Related to the point above, I would have liked to see more ablations on the vote-k selection process. How important is it to use the model confidence “stratification” strategy in the algorithm? This is particularly useful to know since, given a very large set of unlabeled examples `|U|`, it can be costly to run all `u in |U|` model predictions to collect confidence scores.

1. One source of bias that was not addressed in the paper relates to the choice of unlabeled examples (|U|). In this study, the unlabeled corpus for all 10 datasets has been extracted from an existing labeled/supervised corpus (that has been previously annotated). This may not seem like it, but I believe there is huge amount of supervision and work simply *choosing* which examples to label in the first place.  In practice, however, for a new task/dataset, this will not be the case if we want to do only “selective annotation”. I was quite unsure (maybe a little skeptical) that the proposed procedure would work for a “random” or uncurated corpus of unlabeled examples.

1. There is also no discussion on label distribution in the vote-k process. Again, I think the current experimental evaluation is taking advantage of curated training datasets where labels tend to be nicely distributed/represented. So focusing on performing input similarity and model confidence may be enough. For example, for some tasks, a fairly random unlabeled corpus may contain a huge number of “negative” examples (say you want to automatically find pairs of sentences for entailment - most random pairs of sentences will be NEUTRAL). Measuring similarities of inputs alone does not seem to be sufficient to match the expected posterior distribution of labels? Can you please elaborate on this point?

1. With respect to the experimental setup, it is a bit jarring the number of combinations of (dataset, model) pairs (Table 1). I understand that it is expensive/infeasible to run all datasets on all models, and I don’t have a good suggestion to improve on this. But I fear that the end result is that subsequent/future work that builds on vote-k may need to re-evaluate your implementation on a partial set of dataset/tasks.

## Other questions

1. Question about potential clarification in initialization of Algorithm 1. Maybe I did not understand this part of the algorithm, but when running initially |U| contains all examples, and |L| is empty. My understanding is that the score in line 4 of Algorithm 1 (also equation in Section 2.1) generates score of 0 for every node since no u in |U| are yet connected to a node in |L|. Is the argmax using some stable ordering when all scores are 0 to guarantee determinism? Does this choice not influence the final set of examples in |L| ?

1. Recently, the Chain-of-Thought prompting (https://arxiv.org/pdf/2201.11903.pdf) has been used by many papers using in-context learning, with reported improvements in model performance (the paper seems to have been cited already 100 times since its release earlier this year). One question this work does not address is whether model sensitivity/variance to retrieved prompts is also present when used in conjunction with chain-of-thought. And whether the vote-k scheme would be useful to limit the number of chain-of-thought annotations needed to improve in-context learning performance.

**Summary Of The Paper:**

This paper studies NLP tasks in the context of large-language models that are capable of in-context learning. Specifically, the work advocates for constructing prompts dynamically, based on textual similarity with input query, relying on a small pre-annotated corpus. The paper focuses on describing a new technique for selecting good annotation candidates from an unannotated pool of examples to serve as a database for in-context learning. Specifically, they propose a graph-based algorithm, vote-k, which maps unlabeled examples into a similarity graph. The work then describes how retrieval of the examples can be done at inference time to construct prompts for in-context learning. The paper describes several experiments on different large language models (Codex-da-vinci, GPT-J, OPT-175B, etc.) using 10 different datasets. Finally, the paper also compares the vote-k selection strategy with existing selective annotation strategies previously used in active-learning or few-shot learning regimes.


**Summary Of The Review:**

Overall, I think this is a well produced work with clear writing and interesting and insightful experiments. The proposed vote-k algorithm is pretty simple to implement and has potential to be of practical use to users of in-context learning. I do think there are some unanswered questions that I list above, including (1) whether this scheme would be robust to a truly unlabeled or uncurated corpus, and (2) the relationship of vote-k with simpler k-means and other clustering algorithms. Given that these issues do not seem overly problematic, I’m recommending the acceptance of this work.

---

> ### Author Response · Authors · 2022-11-14
> **Thank you for your review**
>
> Thank you for your detailed and thoughtful comments. We are glad to hear that you found the paper very strong and clear with huge potential impacts in broader communities. Below we address the concerns and questions raised in the review. We used purple text to clarify the parts that we modified in the revision.
>
> **Comparisons to clustering algorithms** We agree that comparisons between our vote-k method and clustering algorithms would be useful. Some clustering algorithms, such as agglomerative clustering, require O(n^3) time complexity and are practically inefficient; other algorithms such as k-means clustering, tend to lack diversity, as they always choose the cluster centroid examples. Vote-k first groups examples by uncertainty scores, and then use similarity-based voting to choose the most representative examples from each group. By explicitly emphasizing the diversity and representativeness in selective annotation, every test instance gets a higher chance to find similar examples from the prompt retrieval, which benefits in-context learning during the test time.
>
> **Added k-means results. Underperforms vote-k and lacks diversity** We agree with the reviewer that empirical comparisons with the k-mean baseline would strengthen our results. We added k-mean results to Table 5 in the revision (see the purple text). As seen in the table, k-means underperforms vote-k by 2-3%. We also added k-means to our diversity analysis in Table 10 in Appendix F. This result confirms that k-means clustering chooses less diverse examples compared to our vote-k algorithm.
>
> **Our fast vote-k ablation removes confidence stratification strategy** We share the reviewer’s opinion that it is costly to get model predictions and compute confidence scores. Indeed, we provided an ablation study in Section 4.5 and Table 5: fast vote-k removes confidence prediction in the selective annotation process, and keeps only the similarity-based voting part. As seen in Table 5, this removal of confidence stratification results in 1-2% performance drops with a 10+ times speedup. This means that fast vote-k is an efficient alternative with competitive performance.
>
> **Data selection bias and label distribution** As the reviewer points out, all datasets used in this paper originally come with annotations, and it is possible that some of these datasets are not completely randomly sampled in the construction process. However, we performed analysis on datasets with an imbalanced labeled distribution (e.g., CivilComments in Section 4.3) and datasets sampled **without** any data selection (Civilcomments and Amazon in Section 4.3). Our vote-k algorithm has shown to be effective in these cases as well. Similar to previous work on active learning, we used annotated datasets and removed their annotations in our evaluations. Since we open-source vote-k, we hope that practitioners can use our algorithm for any task of their interest.
>
> **Selective annotation and label distribution** We agree that measuring similarities of inputs alone is not sufficient to match the expected posterior distribution of labels. In practice, however, we often do not know the label distribution on the evaluation/deployment data. For example, the label distribution changes drastically when there are domain shifts (see our experiments on the WILDS benchmark in Section 4.4). We thus believe that selective annotation should not be performed based on label distribution. We add an additional analysis in Appendix H, where we calculate the label imbalance ratio for different selection methods. Across all the selection methods, vote-k indeed results in more balanced label distributions (lower ratio between most frequent class to the least frequent class).
>
> **Not evaluated on all tasks and models** As the reviewer points out, we did not evaluate our algorithm on all tasks with all models. Since we aimed to evaluate our method on a wide range of tasks and datasets from different categories, we chose representative models shown in Table 1. Nonetheless, we used models with different sizes (GPT models from 2.7B to 175B) and different pretraining data (GPT and Codex). We also experimented with 10 diverse datasets with different categories (classification, dialogue, etc.) and different domains. Using our open-source codebase, researchers can use our vote-k algorithm for their customized models and datasets.
>
> **Is initialization in Algorithm 1 deterministic?** Our vote-k algorithm is deterministic even during the first iteration. In Line 4 of Algorithm 1, the score of every unlabeled instance (node) is the sum of s(v) over every incoming edge. As every node got different incoming edges, the first example is deterministically selected with the maximum score.
>
> **Can selective annotation be combined with chain-of-though?** As suggested by the reviewer, Chain-of-Thought prompting can be used together with our selective annotation. We will consider adding these experiments in the final version.

---

> ### Author Response · Authors · 2022-12-04
> **Follow-up with reviewer**
>
> Thanks for your valuable comments. We are writing this follow-up post because it has been days since we posted our initial response. Please let us know if your initial concerns are addressed, and we would appreciate any further questions or comments.

---

### Author Response · Authors · 2022-11-14
**Summary response to all reviewers and the new revision**

We thank all the reviewers for their feedback and constructive comments. We are glad to hear that: our paper is clear and easy to read (Reviewers ZPh1, 3Ufu, bT5a, and dEqn); the experimental section is thorough and the results are strong (Reviewers ZPh1, 3Ufu, and bT5a); our ablation study is detailed and comprehensive (Reviewers 3Ufu and bT5a). We are also pleased that the reviewers think our proposed system is easy to implement with many practical applications (Reviewers ZPh1, 3Ufu, bT5a, and dEqn).

In this work, we aim to propose a new paradigm that is practically useful, e.g., using in-context learning with selective annotation helps to achieve the best performance with the least manual labeling for new, unseen tasks. In the finetuning paradigm, active learning literature has shown that example selection is not very effective for pretrained large language models (discussed in the related work section), while we demonstrated that selective annotation can significantly enhance model performance in the new, in-context learning paradigm.

In the revision, we updated the draft based on the reviewers’ comments. Updates are denoted in purple text for clarity. Our updates are summarized as follows:
* Added results of K-means and Conf-only baselines in Table 5. (Reviewers ZPh1 and dEqn)
* Added results of diversity and representativeness metrics for K-means in Table 10.  (Reviewer ZPh1)
* Added a discussion on the label distribution of selected examples in Appendix H. (Reviewers ZPh1, 3Ufu, and bT5a)
* Added t-SNE visualization in Appendix I and selected examples in Appendix K. (Reviewer 3Ufu)
* We add a discussion in Appendix J on why similar retrieval is necessary for vote-k selection. (Reviewer bT5a)
* Further annotated Algorithm 1 for clarity in Appendix G. (Reviewer dEqn)

---

### Decision · Program_Chairs · 2023-01-20

**Decision:**

Accept: poster

**Justification For Why Not Higher Score:**

The proposed approach is seems straightforward. While it uses ideas and learnings from the previous work and extends them further, the paper may be seen as not too novel.

**Justification For Why Not Lower Score:**

The proposed approach is interesting and useful, and the paper includes detailed experimentation on a large set of tasks, and will be useful for many practitioners.

**Metareview: Summary, Strengths And Weaknesses:**

In-context learning, where a few examplars are provided to large language models as additional input demonstrations during inference time has been shown to be useful for natural language processing tasks. This paper proposes to selectively sample a pool of unlabeled examples for annotation, to be used as in-context examplars in advance. At inference time, a subset of these annotated examples are retrieved for use for each test example. A set of methods inspired by previous work on active learning has been used for selective sampling. Experimental results on 10 datasets, including tasks like common sense reasoning (HellaSwag) and dialog state tracking, has shown significant improvements in the amount of data annotation needed.
The contents of the paper are timely and useful, and seem easy to implement and execute. The paper includes detailed experimentation on a large set of tasks. The proposed method is a bit simplistic. Reviewers made a good set of suggestions regarding the weaknesses of the paper, including lack of some experimental results. Authors have included additional experimental results (such as comparison of vote-k to k-means clustering) after the rebuttal to motogate these weaknesses.

**Note From Pc:**

if the above contains the word "oral" or "spotlight" please see: "oral" presentation means -> notable-top-5% and "spotlight" means -> notable-top-25%. As stated in our emails, we are disassociating presentation type from AC recommendations